# Improved Cathepsin Probes for Sensitive Molecular Imaging

**DOI:** 10.3390/molecules27030842

**Published:** 2022-01-27

**Authors:** Yonit Yitzhak, Hanmant Gaikwad, Tommy Weiss-Sadan, Emmanuelle Merquiol, Boris Turk, Galia Blum

**Affiliations:** 1The Institute for Drug Research, The School of Pharmacy, Faculty of Medicine, The Hebrew University of Jerusalem, Jerusalem 91120, Israel; yonityi@post.jce.ac.il (Y.Y.); hanmant.gaikwad@cuanschutz.edu (H.G.); tweisssadan@mgh.harvard.edu (T.W.-S.); emmanuellem@savion.huji.ac.il (E.M.); 2Department of Biochemistry and Molecular Biology, J. Stefan Institute, 1000 Ljubljana, Slovenia; boris.turk@ijs.si; 3The Wohl Institute for Translational Medicine, Hadassah Hebrew University Medical Center, Jerusalem 91120, Israel

**Keywords:** cathepsin, activity-based probes, cell permeability

## Abstract

Cysteine cathepsin proteases are found under normal conditions in the lysosomal compartments of cells, where they play pivotal roles in a variety of cellular processes such as protein and lipid metabolism, autophagy, antigen presentation, and cell growth and proliferation. As a consequence, aberrant localization and activity contribute to several pathologic conditions such as a variety of malignancies, cardiovascular diseases, osteoporosis, and other diseases. Hence, there is a resurgence of interest to expand the toolkit to monitor intracellular cathepsin activity and better ascertain their functions under these circumstances. Previous fluorescent activity-based probes (ABPs) that target cathepsins B, L, and S enabled detection of their activity in intact cells as well as non-invasive detection in animal disease models. However, their binding potency is suboptimal compared to the cathepsin inhibitor on which they were based, as the P1 positive charge was capped by a reporter tag. Here, we show the development of an improved cathepsin ABP that has a P1 positive charge by linking the tag on an additional amino acid at the end of the probe. While enhancing potency towards recombinant cathepsins, the new probe had reduced cell permeability due to additional peptide bonds. At a second phase, the probe was trimmed; the fluorophore was linked to an extended carbobenzoxy moiety, leading to enhanced cell permeability and superb detection of cathepsin activity in intact cells. In conclusion, this work introduces a prototype design for the next generation of highly sensitive ABPs that have excellent detection of cellular cathepsin activity.

## 1. Introduction

Cysteine cathepsins (CTSs) are proteolytic enzymes that are involved in a variety of physiologic processes. These include housekeeping activities such as autophagy and transcriptional control, but numerous pieces of evidence also implicate them in different pathologic conditions such as inflammatory diseases and cancer [1,2,3]. 

Members of the cathepsin family, CTS B, L, and S, were believed to require acidic conditions such as in the lysosomes [2,3,4]. More recently, however, this view was refuted, as CTSs were also evident in unexpected cellular compartments such as the cytosol, nucleus, and extracellular milieu [1,4,5,6,7].

While extracellular CTS activity is extensively documented in contexts such as cancer metastasis or inflammatory conditions, where CTSs are over-expressed, there are still many neglected features that remain to be uncovered [8]. These, however, are hindered due to the lack of sensitive tools to monitor CTS activity. Such activities include nuclear cathepsin function during the cell cycle or under conditions of cellular starvation. Tools that report on activity are needed to investigate CTS involvement in these activities.

The two major groups of tools used to monitor protease activity are substrate probes and activity-based probes (ABPs) [9]. Substrate probes are usually comprised of substrate sequences that generate a fluorescent signal after cleavage. Their signal is produced by separation of a fluorophore-quencher pair, or by separation of two fluorophores that auto-quench, each attached at a different side of the cleavage point. ABPs are probes that covalently label active enzymes. They are usually small molecules that comprise a recognition sequence, a tag for detection, and an electrophile “warhead” that enables the covalent attachment to the active site residue [10].

Here, we set out to generate a cell permeable highly sensitive ABP that will report on CTS B, L, and S activities in live cells. We based our design on GB123, a fluorescent ABP that enables detection of the activity of these enzymes in intact cells and in vivo [11,12]. A positive charge at the P1 position is known to be preferred by CTS B, L, and S. Nevertheless, the P1 position was originally used to link the fluorescent tag in GB123—blocking the positive charge of the probe and introducing a bulky group close to the cleavage site. Thus, to increase the potency of the probe, we designed probes that have a positive charge at the P1 position by extending the probe and attaching the tag at the P4 position. This modification, while elevating the potency, resulted in reduced cell permeability as the probe was extended to link the fluorophore. Therefore, at a second step, the probe was trimmed and further modified for tag attachment, resulting in an improved cell permeable cathepsin probe that enables superb cathepsin detection.

## 2. Results

### 2.1. Probe Design 

In an attempt to improve the potency of our published cathepsin ABP, GB123, we generated a probe that has a positive charge at the P1 position. Initially, we synthesized YHY1 where we switched the CBZ capping group of GB123 with a phenylalanine moiety and further extended the peptide portion with a lysine (at the P4 position) to attach the fluorophore. In addition, to increase cell permeability, the Cy5 fluorophore of GB123 was replaced with a BODIPY TMR-X fluorophore, Figure 1. We also generated control compounds, YHY2-5, to investigate the contribution of the P1 positive charge, probe sequence, and fluorophore position to the potency of the probe. The four YHY1 control probes include YHY2, composed of the same scaffold but with the fluorophore attached to the P1 (thus omitting its positive charge), and a free amine at P4, YHY3, with the same scaffold but with two fluorophores on P1 and P4, lacking positive charges. In addition, YHY4, a shorter probe lacking the P4 lysine and with the P1 linked to the fluorophore, was included and YHY5, a probe with the scaffold of GB123 but bearing a BODIPY TMR-X fluorophore instead of Cy5. Compounds were generated with 1–6% overall yield and a purity of above 95% (YHY2, 88%); the synthesis of all compounds is described in Appendix A.

### 2.2. Biological Evaluations of Probes

In order to investigate if the affinity toward cathepsins B and L is increased by a positively charged P_1_ residue, the five probes YHY1–5 were tested for their binding and labeling of recombinant cathepsin B and L. The same probes were then evaluated for cell permeability and labeling of endogenous cathepsins within intact NIH 3T3 cells. 

### 2.3. Labeling of Recombinant Cathepsin B and L

Cathepsin B and L were pretreated with or without a cathepsin inhibitor GB111-NH_2_ [12] for 30 min. Enzymes were then labeled by a one-hour incubation with increasing concentrations of probes YHY1–5 (for Cat L only, YHY1–4 were used), and samples were then resolved by SDS PAGE. Potency towards intact cathepsins was determined by the intensity of the fluorescent probe-enzyme complex detected between 25 and 37 kDa. YHY1 labeled both cathepsin B and cathepsin L of markedly higher potency than probes YHY2–4, Figure 1. This is very pronounced in cathepsin L when comparing YHY1 to YHY2 in that both have the same peptide sequence, and only the positive charge is at a different position. Therefore, we concluded that the positive P1 charge indeed increases potency towards both cathepsin B and L.

### 2.4. Labeling of Cathepsins in Intact Cells 

To test cell permeability, intact NIH-3T3 cells were used in culture. Cells were pretreated with a vehicle or GB111-NH_2_; then, the ABPs were added to the growth media of the living cultures, and after six hours the cells were collected, lysed, and resolved by SDS PAGE. Cell permeability and potency of labeling intact cathepsins were determined by the intensity of the fluorescent probe-cathepsin adduct seen between 20 and 37 kDa in Figure 2. As expected, YHY1 was more potent than YHY2–4; however, YHY5, which was less potent than YHY1 in labeling recombinant cathepsin B, was found to label both endogenous cathepsin B and L more potently. We concluded that poor cell permeability is limiting YHY1 from efficiently penetrating the cells and labeling the endogenous cathepsin proteases. In addition, YHY4 and YHY5 that label cathepsin B in similar potency had dramatically different labeling in intact cells. Albeit both YHY4 and YHY5 have the fluorescent tag at the same position, they differ at the P3 position; YHY5 has a carbobenzoxy (CBZ) moiety, while YHY4 has an acetylated phenylalanine. We concluded that the additional amide bonds in YHY1–4 limit their cell permeability.

### 2.5. Second Generation Probes 

To increase cell permeability, we generated a new scaffold where we maintained the CBZ group and linked the fluorophore directly to it, maintaining the positive charge at P1; see Figure 2. This scaffold was then labeled with either BODIPY TMR-X (HG122) or Cy5 (HG121). The probes were compared with the published GB123 [11], also bearing a Cy5 fluorophore. The structures and synthesis of the second-generation probes are described in the Appendix A.

### 2.6. Biochemical Evaluation of the Second Generation Probes 

After the synthesis of the second-generation probes, the BODIPY TMR-X probe, HG122 was evaluated biochemically. Labeling of both recombinant cathepsin B and endogenous cathepsins in intact NIH-3T3 cells was compared to the published GB111-TMR-X [12] and the first-generation probe YHY-1, Figure 3a,c. HG122 labeled recombinant cathepsin B significantly better than both YHY-1 and GB111-TMR-X. A similar increase in cathepsin B labeling was detected in intact cells. Furthermore, the Cy5 labeled probe HG121 was also significantly more efficient than the published GB123 in labeling both recombinant and endogenous cathepsin B, Figure 3b,d.

## 3. Discussion

This manuscript describes our success in increasing the potency of fluorescent activity-based probes for cathepsins by exploiting the active site negative charge at the S1 binding pocket by designing novel probes that have reduced bulkiness and a positive P1 residue. While this strategy proved to be useful, the extra amino acid and amide bonds that were introduced to link the fluorophore to the peptide rendered the probe less cell permeable, resulting in reduced cathepsin labeling in intact cells. We thus devised a chemical strategy to extend the CBZ group and linked the fluorophore to it. These manipulations resulted in two probes with a similar structure to the original GB123 probe but with a free amine at the P1 position where the fluorophore was linked to a modified CBZ. 

In this work, we touch upon the delicate balance between potency and cell penetrance of chemical tools. While the free positive charge increases both cathepsin B and cathepsin L binding, for practical use, the elimination of the extra amide bonds resulted in two highly potent novel probes HG121 and HG122. HG121 could serve as an excellent chemical tool for detection of cathepsin activity in live cells as well as in vivo as HG121 is labeled with a near IR fluorophore, Cy5. Thus, it could serve the scientific community for improved non-invasive cancer detection as was shown with other fluorescent cathepsin ABPs. 

## 4. Materials and Methods

### 4.1. General Methods

Unless otherwise noted, all resins and reagents were purchased from commercial suppliers and used without further purification. All amino acids (99% purity) and resin were purchased from GL Biochem, Shanghai China. BODIPY TMR-X, SE was purchased from Invitrogen, Waltham, Massachusetts USA. Solvents and reagents were purchased from Bio-Lab and Sigma-Aldrich. All solvents used were HPLC grade. All water-sensitive reactions were performed in anhydrous solvents under a positive pressure of argon. All light-sensitive reactions were performed in the dark. Reactions were analyzed by Liquid Chromatography Mass Spectrometry (LC-MS) with a single quadrupole mass spectrometer (Thermo Scientific MSQ Plus, Waltham, MA, USA), attached to an Accela UPLC system. All final compounds were purified by reversed-phase HPLC in acetonitrile/water (ACN/DDW) gradient, using the DIONEX UltiMate 3000 system (Thermo Scientific, Waltham, MA, USA). Recombinant cathepsin B and L were supplied by B. Turk (J. Stefan Institute). Fluorescent gels were scanned with a Typhoon 9400 flatbed laser scanner (GE Healthcare, North Richland, TX, USA).

### 4.2. Chemical Synthesis and Characterization of Probes

In general, synthesis of each of the seven ABPs (YHY1-5 and HG121-122) was conducted by coupling two separate parts; this is due to the presence of two lysines (Lys) in the probes and to increase the yields see Appendix A. For example, to generate the YHY1, first, the Aceto-Lys-(Boc)-Phe-Phe-COOH, (Aceto-Lysine (Boc)- Phenylalanine- Phenylalanine- COOH), which serves as part of the recognition element by the CTSs, was synthesized on a solid support. Fmoc-Lys-DMBA-AOMK (Fmoc-Lysine-dimethyl benzoyl- acyloxy methyl ketone), where the AOMK serves as an electrophilic warhead, was synthesized in solution and attached to the 2-Chlorotrityl chloride resin. Following, the Aceto-Lys-(Boc)-Phe-Phe-COOH was coupled to the resin anchored (Fmoc)-Lys-DMBA-AOMK. The last step was removing the Boc protecting group and attaching a fluorophore, BODIPY TMR-X, to the Aceto-Lys-(NH_2_)-Phe-Phe-Lys-DMBA-AOMK that was then cleaved, yielding YHY1 ABP. YHY1 was purified by semi-preparative HPLC (reversed-phase), and purity was determined using LC-MS (reversed-phase). Finally, the probe was lyophilized to obtain a dark purple solid. The syntheses of the other YHY2–5 and HG121-122 ABPs were conducted similarly, except for some changes, as described in the results. 

### 4.3. Solid-Phase Peptide Synthesis

2-Chlorotrityl chloride resin was loaded by shaking the resin for 2 h with 3 equivalents (eq.) of an amino acid and 3 eq. of dried N-methyl diisopropylamine (DIEA) dissolved in anhydrous dimethyl formamide (DMF). The resin was capped by shaking with methanol (1 mL/g resin) for 20 min. Resin load was determined by adding 3 mL of 20% piperidine solution (in DMF) into two test tubes containing 1mg dried resin, for 15 min and shaking, then reading the absorption of each tube at 290 nm, using the 20% piperidine solution as blank. 

### 4.4. Removal of Protecting Groups

The Fmoc protecting group was removed by incubation with 20% piperidine/DMF (*v*/*v*) for 2 × 15 min followed by DMF and dichloromethane (DCM) washes. When AOMK was present, to avoid its hydrolysis, the Fmoc was removed with 0.1 M tetra-n-butylammonium fluoride (tBaf) in dry tetrahydrofuran (THF)/ dry DMF (*v*/*v*, 50:50) instead. The Boc protecting group was removed by incubation with 25% Trifluoroacetic acid TFA/anhydrous DCM (*v*/*v*) for 20 min followed by co-evaporation with toluene 4 times. When the Boc was removed from a peptide on the resin (i.e., for YHY1 synthesis), the peptide was incubated with 4 M HCl in dioxane [13] for 2 × 15 min followed by DMF and DCM washes. 

### 4.5. Compound Cleavage from Resin

Peptides were cleaved from the resin by 3% or 10% TFA/DCM (*v*/*v*), when the Boc protecting group was present, or absent, respectively. The cleavage solution was collected, and the solvent was removed by co-evaporation with toluene. The crude peptide was then purified by semi-preparative HPLC (reversed-phase) or used without further purification. The product was dried in vacuo, and purity was determined by LC-MS using a reversed phase column.

### 4.6. Purification and Characterization

All final products and most semifinal products were purified by reversed-phase semi preparative HPLC with an ACN/DDW gradient (with 0.1% TFA), using the DIONEX UltiMate 3000 system (Thermo Scientific, Waltham, MA, USA). The system was equipped with a Waters WAT082887 (8 × 100 mm) C_4_ column, (5 mL total volume), or a Waters preparative LC 25 × 100 mm Module C_18_ column (49 mL total volume). Purification was conducted using a flow of 4 or 15 mL/min for a 5mL or 49 mL column, respectively. All products were analyzed by LC-MS with a single quadrupole mass spectrometer (Thermo Scientific MSQ Plus, Waltham, Massachusetts USA) attached to an Accela UPLC system. LC-MS chromatography was conducted with an ACN/DDW gradient (with 0.1% Formic acid), a flow of 600 µL/min using a Phenomenex Onyx Monolithic C_18_ (size 2.0 × 50 mm) or Waters Symmetry300 C_4_ (particle size 5 µm, size 4.6 × 250 mm, product of USA) column. 

### 4.7. Recombinant Cathepsin Labeling of the Probes

After probes’ synthesis and chemical evaluation, the affinity to recombinant cathepsin B or L was tested in acetate buffer (50 mM acetate, 4 mM DTT, and 5 mM MgCl_2_, pH 5.5); enzymes were pre-treated with or without a cathepsin inhibitor, GB111-NH_2_ [12,14] for 30 min (indicated samples) at 37 °C. Increasing concentrations of the ABPs were added to samples for 60 min. The reaction was stopped by the addition of 4X sample buffer (40% glycerol, 0.2M Tris/HCl 6.8, 20% β-mercaptoethanol, 12% SDS, and 0.4 mg mL^−1^ bromophenol blue). Samples were resolved by a 12.5% SDS gel and scanned by a Typhoon laser flatbed scanner at 532/580 nm. For Cy5 labeled probes (HG122 and GB123), gels were scanned at 633/675 nm.

### 4.8. Cell Permeability Assay

Probes were evaluated for cell permeability in intact NIH-3T3 cells. Cells were seeded in 12-well plates one day before treatment. Cells were pretreated with the cathepsin inhibitor GB111-NH_2_ (2 µM) or with control DMSO (0.1%) for 1 h and then labeled by the addition of the probes (at indicated concentrations) in growth medium DMSO (0.1%) for 6 h; the final DMSO concentration was maintained at 0.2%. Cells were then washed with phosphate-buffered saline (PBS) and lysed with RIPA buffer (PBS pH 7.4, 1% Nonidet P-40, 0.5% Sodium deoxycholate, 0.1% SDS). Crude lysates were spun down. Equal amounts of protein (40 µg per lane) were resolved by a 12.5% SDS-PAGE, and labeled proteases were visualized by scanning of the gel with a Typhoon flatbed laser scanner (excitation/emission 532/580 nm or 633/675 nm). 

### 4.9. Gel Quantification and Statistical Analysis

Quantitative analyses of the scanned gels were performed using the Image J 1.51d (NIH) software [15], and statistical analysis was performed using Microsoft excel. Data is presented as the mean ± SD of 2 or 3 technical replicates per condition. Means of data were compared using an unpaired, two-tailed *t*-test. A *p* ≤ 0.05 was considered statistically significant. 

Detailed methods of the synthesis of all compounds are described in Appendix A also using methods described in [16,17,18,19,20].

## 5. Conclusions

Changes in structure have a direct influence on both potency and cell permeability of chemical probes. Positioning the fluorophore as an extension of the CBZ moiety at P3, away from the cleavage site, resulted in a highly potent and cell permeable probe that labels cathepsin B and L efficiently in intact cells.

## Data Availability

The data presented in this study are available in Appendix A.

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
