# Peer review of "Improved Cathepsin Probes for Sensitive Molecular Imaging"

_molecules, 2022, doi:10.3390/molecules27030842_

Round 1

Reviewer 1 Report

This is a nice study that has ultimately resulted in a non-quenched cysteine cathepsin probe with improved efficacy compared to its predecessor. Specifically, the authors examined the influence of the positive charge and placement of the fluorophore on the P1 sidechain on probe efficacy. They found that probes with an uncapped lysine (free amine) at P1 exhibited improved efficacy compared to probes in which the fluorophore was attached via the side chain of a lysine in the P4 position.

The text suggests that this difference is attributable to the presence of the free amine; however, the influence of the free amine and the fluorophore cannot be uncoupled by these data. It could be the amine, or it could be the reduced bulk in the P1 position that drives the increase in potency. To directly test this, probes in which a smaller less obtrusive capping group on the lysine side chain would be required. In lieu of additional experiments, I would suggest revising the text to not overinterpret the available data. Alternatively, some molecular modelling could shed some light on this question.   

In the second probe series, the authors attempt to improve permeability of the optimised probes by shortening the peptide. This appears to have been very successful for the Cy5-tagged probe, but far less so for the BODIPY-TMRX probe. Could the authors comment on this discrepancy in more detail?

Moreover, in this experiment, it is difficult to uncouple potency/affinity from permeability. As such, it would perhaps be useful to see the potency of the HG probes compared to GB123/GB111 against recombinant cathepsins (as was done for the YHY series). Is HG121 better than GB123 because it is more potent, more permeable or both?

Lastly, the introduction specifies the need for new cathepsin probes that can measure enzymatic activity in real time. This probe series is non-quenched, which means that extensive washout of unbound probe is required before imaging. Would a quenched analogue of HG121 be feasible or would the increased distance between the fluorophore and quencher reduce quenching efficiency? Perhaps the authors could comment on this in the discussion.

Author Response

We thank the reviewer for the constructive comments!!

Reviewer 2 Report

Yitzhak and colleagues report on the development of new cathepsin probes, and suggest that these would be highly sensitive in detecting activity of cathepsin B, S and L.

It is unclear to me to what extent these new probes are performing better than the already published ones. In binding experiments, one would probably like to see side by side comparison with available published compounds.

What about cellular toxicity over time?

What about specificity? Other proteases/cathepsins to be targeted? Whole gels are missing to appreciate this, additional western blotting would be required to determine specificity.

Fluorescence analysis could be used to show cell permeability

Author Response

(The authors gave the same response as above.)

Reviewer 3 Report

In the submitted manuscript, the laboratory of Galia Blum reports on new, improved probes for cysteine cathepsins. In previous designs, the fluorophore in acyloxymethyl ketone-based activity-based probes (ABPs), was placed at the P1 Lysine position. I never really understood why this initial design was taken, as it places a very bulky fluorophore near the active site. In addition, masking the positive side chain of the P1 lysine severely reduces the potency of these type of probes.

In this paper, Blum and co-workers reveal that extension of the peptide scaffold and placement of the fluorophore at the N-terminus leads to poor cell permeability because of the extra amino acids/amide bonds. To circumvent this problem, they report on a probe with a Cbz-like structure that can be linked to a fluorophore. This probe turns out to be equal to superior to the original GB123 probe, which bears the fluorophore in the P1 position.

Overall, this paper reports a new type of cathepsin ABP, which may be the basis for improved future ABPs for cysteine cathepsins and their downstream applications.

In view of this, I recommend publication of this manuscript after taking into consideration the following minor comments.

  1. Line 44: fucntion should read: function
  2. Line 47: The substrate probes -> Substrate probes
  3. Line 47/48: a substrate sequence that after cleavage generate -> this sentence doesn’t read well because of a discrepancy in singular/plural. Better: Substrate probes are usually comprised of substrate sequences that generate a… etc
  4. Line 50: The ABPs -> ABPs
  5. Line 51: ‘as a result of enzymatic activity’ may be deleted
  6. Line 51: the sentence seems to be 2 sentences. Insert a ful stop after ‘enzymes’ and start the following one with “They are usually small molecules…”
  7. Overall, I recommend reading over the manuscript (perhaps by a native speaker) to check for typos or grammar mistakes.
  8. Line 66: activity-based probes -> replace by ABP (abbreviation is already given before)
  9. Line 79: 11 can be deleted (main text doesn’t contain probes with Arabic numbers, just compound names YHY)
  10. Line 93: ‘spread by SDS PAGE’ -> separated/resolved by SDS PAGE. Same in line 109
  11. Line 133/134: compound names should go in brackets (HG122/121)
  12. Line 134: ‘that was compared…’ grammatically incorrect. Better full top after HG121, then: “The probes were compared with the published…”
  13. Scheme 2: why is the Phe in the P2 position sketched as a racemized structure? Is this a sketching error or did it racemize during coupling (I wouldn’t expect so, because the nitrogen is carbamate protected, which usually prevents racemization)
  14. Line 139/140: should be part of the figure caption
  15. Figure 3: GB111 -> should be GB111-TMRX
  16. Line 149-152: should be part of the figure caption
  17. Line 157: should read: ‘the extra amino acid and amide bond that were introduced’ (typo in amid + plural should be used)
  18. Line 179: “in dark” -> “in the dark”, Mass spectrometer -> mass spectrometry
  19. Line 183: recombinant… etc: a verb is missing
  20. Line 248: a strange sign instead of micro/Greek letter mu is depicted.

Author Response

We would like to thank the reviewer very much for the thorough review. All edits were implemented in the text.

Round 2

Reviewer 1 Report

Overall, the author's have sufficiently addressed my queries in the response to the reviewer.

In addition to the introduction/discussion, I suggest the text in the results section be modified to reflect that the difference in potency could be due to either the positive charge or bulk of the fluorophore. Line 75-6 "We also generated control compounds, YHY2-5, to investigate the contribution of the P1 positive charge to the potency 76 of the probe" The controls do not independently test for the positive charge, so just make this more explicit.

The addition of the figure in the supplemental section does help to clarify that the differences are due to improved permeability rather than potency. However, these data were not referred to or discussed in the main manuscript. As there are no other results in the supplemental section, I would recommend incorporating the gels into Fig 3 and describing them in the text. You could add the quantification as you did for the other gels. One other important comment - it appears to my eye two gels are spliced together to make both the top and bottom panel (sharp contrast between the two halves and the left and right side are slightly different sizes or not quite aligned perfectly). I suspect what has happened is that other samples run on the same gel were excised. This is fine, but it is best practice to separate the gels - showing clearly where it was cropped - and indicate in the legend that the samples were cropped from the same gel and that the gain settings were adjusted in the same way for each panel. This would remove any and all doubt about image manipulation. 

Author Response

We thank the reviewer very much for the excellent comments that improved the manuscript even more!!

Reviewer 2 Report

The authors commented on my questions reasonably. One key point to me seems though to be toxicity/effect on cell numbers. this is key also to evaluate specificity and sensitivity. how do the authors make sure that they compare equal cell numbers and equal protein (details are missing here). With respect to cell permeability answer on my last comment earlier), I thought this was the whole point, to generate cell permeable probes??

One minor point, what conditions does the quantification in Fig 3c and 3d relate to exactly?

Author Response

We thank the reviewer for the constructive comments! Please see the attachment.
